# Nanoparticles for Mitigation of Harmful Cyanobacterial Blooms

**DOI:** 10.3390/toxins16010041

**Published:** 2024-01-12

**Authors:** Ilana N. Tseytlin, Anna K. Antrim, Ping Gong

**Affiliations:** 1Oak Ridge Institute for Science and Education, 1299 Bethel Valley Road, Oak Ridge, TN 37830, USA; ilanatse@pitt.edu; 2School of Pharmacy, University of Pittsburgh, 3501 Terrace St., Pittsburgh, PA 15261, USA; 3Environmental Laboratory, U.S. Army Engineer Research and Development Center, 3909 Halls Ferry Road, Vicksburg, MS 39180, USA; anna.k.antrim@erdc.dren.mil

**Keywords:** metal/carbon/organic-based nanoparticles, manufactured/natural nanoparticles, harmful cyanobacterial bloom (HCB), cyanotoxin, eutrophication, mitigation, photocatalysis, cytotoxicity, reactive oxygen species (ROS), flocculation/coagulation, nutrient (phosphorus) removal, adsorption

## Abstract

With the rapid advancement of nanotechnology and its widespread applications, increasing amounts of manufactured and natural nanoparticles (NPs) have been tested for their potential utilization in treating harmful cyanobacterial blooms (HCBs). NPs can be used as a photocatalyst, algaecide, adsorbent, flocculant, or coagulant. The primary mechanisms explored for NPs to mitigate HCBs include photocatalysis, metal ion-induced cytotoxicity, physical disruption of the cell membrane, light-shielding, flocculation/coagulation/sedimentation of cyanobacterial cells, and the removal of phosphorus (P) and cyanotoxins from bloom water by adsorption. As an emerging and promising chemical/physical approach for HCB mitigation, versatile NP-based technologies offer great advantages, such as being environmentally benign, cost-effective, highly efficient, recyclable, and adaptable. The challenges we face include cost reduction, scalability, and impacts on non-target species co-inhabiting in the same environment. Further efforts are required to scale up to real-world operations through developing more efficient, recoverable, reusable, and deployable NP-based lattices or materials that are adaptable to bloom events in different water bodies of different sizes, such as reservoirs, lakes, rivers, and marine environments.

## 1. Introduction

As the largest, most diverse, and most widely distributed group of photosynthetic organisms, cyanobacteria are an important primary producer involved in many biogeochemical processes and form part of the phytoplankton in aquatic ecosystems [1]. Cyanobacteria are also known as blue-green algae, but strictly speaking, they are not algae, a name reserved for eukaryotic phototrophs [2]. Due to climate warming and eutrophication, harmful cyanobacteria blooms (HCBs) have not only increased in frequency, magnitude, and duration worldwide in the past several decades but also posed serious threats to ecological diversity, environment quality, human health, aquaculture, fisheries, recreation, and tourism [2,3,4].

There is a plethora of evidence linking excess nutrient (e.g., N and P) loading and warm water temperatures with stimulated growth of both pelagic and benthic cyanobacteria [5,6,7,8]. Through regulating their buoyancy to occupy the optimal niche of light and nutrient availability [9], massive growth of pelagic cyanobacteria increases turbidity, restricts light penetration, and reduces the amount of photosynthetically available radiation in the water column for other photosynthetic aquatic biota (e.g., algae, phytoplankton, and macrophytes) [6,10]. HCBs also cause hypoxia when the overgrown cyanobacteria, algae, and phytoplankton die and sink to the lake bottom or sea floor, where microbial decomposition depletes the oxygen, leading to the suppression of the growth of benthic species [11]. Hence, HCBs negatively impact aquatic ecosystem diversity.

Furthermore, some bloom-forming genera (e.g., *Microcystis*, *Anabaena* (now *Dolichospermum*), *Aphanizomenon*, *Cylindrospermopsis*, *Lyngbya*, *Nostoc,* and *Planktothrix*) can produce a variety of cyanotoxins, such as microcystins, anatoxins, cylindrospermopsins, and saxitoxins [12]. These cyanotoxins and other secreted harmful metabolites not only deteriorate water quality and threaten shellfish and other grazing zooplankton but also cause human illness and poisoning of shore birds and aquatic carnivores via the food chain [13]. In addition, a 10% increase in HCB frequency reduced the average home values by 3.3~4.3% for properties near > 2000 US inland lakes during the years 2008–2011 [14].

## 2. Existing Strategies for HCB Mitigation

The ideal strategy is to control the root causes of HCBs, that is, global warming and eutrophication. However, to get these under control requires long-term and consorted efforts worldwide or at least on a regional scale, for example, reducing nutrient input from external sources into aquatic ecosystems. Current strategies focus on short-term and local solutions. To prevent or mitigate the adverse effects of HCBs, tremendous amounts of R&D efforts have been made in developing a wide variety of technologies and approaches, which can be classified based on different criteria. For instance, they may be categorized into direct (i.e., removal and proliferation inhibition of bloom-forming cyanobacteria) and indirect control (i.e., nutrient input/sink or precipitation/sequestration control) by treatment subject, or physical/mechanical, chemical, and biological methods by treatment mechanism, or spot, water body (partial or whole) and watershed control by treatment area, or water column and benthic/sediment control by treatment layer/vertical depth. Although there were five critical reviews recently published on this topic (see [1,15,16,17,18]), each review had limitations and lacked comprehensive coverage of their respective subject area. Here, we attempt to integrate the contents of these five review articles and fill the missing gaps with some additional literature. Unless specified, these five reviews and references therein are not cited in this section. Our intention is to provide a brief overview without going into details or specificities to avoid repetition. This section serves as the background for our focused review of a specific group of emerging nanotechnologies for HCB mitigation in the following sections.

### 2.1. Physical or Mechanical Methods

There has been a wide array of physical or mechanical methods developed to mitigate HCB effects, including artificial mixing or water column circulation, hypolimnetic (deep-water) oxygenation or aeration, dredging [17], sonication [1,17], mechanical harvesting [1], magnetic separation, centrifugal separation, and ultraviolet (UV) radiation [18].
Artificial mixing involves the manipulation of water circulation within a lake or reservoir to weaken or eradicate the density stratification of the water column.Hypolimnetic oxygenation or aeration treatments can maintain or increase the dissolved oxygen (DO) level and meet oxygen demand in the anoxic hypolimnion without disrupting thermal stratification.Dredging is a geoengineering technique that excavates sediments in surface water sources and relocates the sediments to a disposal site, leading to the removal of nutrients from their sink (sediment).Sonication (ultrasonic radiation) applies high-frequency (>20 KHz) ultrasound to generate a cyclic expansion and compression phase, leading to the disruption and collapse of gas vacuoles responsible for regulating buoyancy in cyanobacteria cells. Such acoustic cavitation results in sedimentation and subsequent photosynthesis inhibition of the floating cells (e.g., *Microcystis aeruginosa*, *Synechococcus* sp., and *Anabaena circinalis*).Mechanical harvesting can remove the accumulated biomass of toxic cyanobacterial cells (e.g., pelagic colony-forming *Microcystis* spp. and benthic mat-forming *Microseira wollei*).Magnetic separation uses recyclable natural magnetic sphalerite (NMS, a naturally occurring and earth-abundant mineral [19]) or a mixture of iron oxide and chloride powder to adsorb and disrupt bloom plankton (including cyanobacteria) through physical interactions and a magnetic separator to remove the adsorbed plankton from the water column.Centrifugal separation works by pumping bloom water through a centrifugal separator to segregate cyanobacteria in the bloom water.UV radiation can induce drastic damage to the thylakoid, a membrane-bound photosynthesis compartment inside cyanobacteria, leading to cell death. For example, a 6 h treatment of 11.8 W/m^2^ UV-A (315–400 nm) caused 90% mortality in *Cylindrospermopsis raciborskii*, a filamentous nitrogen-fixing cyanobacterium [20].


Physical/mechanical methods take effect quickly, often within hours or even minutes (e.g., sonication and UV radiation), and leave no or little environmental residue. However, they are also very energy-intensive, have low scalability, and have low target-specificity.

### 2.2. Chemical Methods

Chemical treatment is currently the most promising and well-established strategy for HCB mitigation. It includes two main groups of approaches: algaecide for cyanobacterial biomass suppression and coagulants/flocculants for nutrient sequestration [16].

The following chemicals have been used as algaecides: herbicides such as Diuron (3-(3,4-dichlorophenyl)-1,1-dimethylurea) and Endothall (7-oxabicyclo [2.2.1]heptane-2,3-dicarboxylic acid), copper (Cu), potassium permanganate (KMnO_4_), hydrogen peroxide (H_2_O_2_), ozone (O_3_), and oxygen- or ozone-infused micro- or nanobubbles. In addition, cationic peptides (e.g., TD53, HPA3, and HPA3NT3) and some synthetic or microbe-produced surfactants such as cocamidopropyl betaine, ethylene bis(dodecyl dimethyl ammonium bromide), sophorolipids, and rhamnolipids exhibited strong algicidal efficacy [18].

The following materials have been used as coagulants or flocculants in a lake or reservoir to precipitate or bind excess phosphorous (P): aluminum (in the form of Al salts such as Al_2_(SO_4_)_3_, AlCl_3_, and poly-aluminum chloride (PACI)), iron (FeCl_3_, FeCl_2_, and Fe(SO_4_)_3_), calcium (lime (CaO, CaCO_3_, and Ca(OH)_2_) or calcite (CaCO_3_)), and modified clay particles (e.g., lanthanum modified bentonite). Some coagulants can also bind cyanobacterial cells, resulting in their sedimentation.

The above two approaches may complement each other because algaecides can rapidly remove cyanobacteria temporarily without reducing the aqueous or sediment-bound nutrient concentrations, which can be achieved by coagulants and flocculants. For instance, clay particles modified with the abovementioned (bio-)surfactants can improve both biomass removal rates and reduce the amount of required clay by an order of magnitude [18]. Although chemical methods provide a cost-effective and fast solution for HCB mitigation, their ecological impacts on non-target aquatic flora and fauna need to be carefully evaluated.

### 2.3. Biological Methods

Many biological resources have been explored as agents to combat HCBs. They range from cyanophage (viruses that specifically target cyanobacteria), bacteria, and fungi, to algae and macrophytes, and to zooplankton and fish. These organisms can directly ingest/graze (e.g., water flea, mussel, and bighead carp), lyse/attack (e.g., lytic cyanophage, algicidal bacteria/actinomycetes, and parasitic *Amoebophrya* sp.) cyanobacteria, or indirectly inhibit the growth and reproduction of cyanobacteria through the release of allelopathic compounds (e.g., macrophytes, barley, and rice straws). The most promising biological methods include cyanophage treatment, biomanipulation, and allelopathy. Despite many biotic and abiotic factors that may influence life cycle, infectivity, efficacy, and scalability, cyanophages have the potential to provide a highly specific control strategy with minimal impacts on non-target species and propagation in the environment [21]. Biomanipulation involves increasing the pressure on phytoplankton communities by reducing or removing planktivorous fish or by increasing grazer and zooplankton populations. Allelopathy is a biological phenomenon by which an organism (e.g., plants, bacteria, coral, and fungi) produces allelopathic biochemicals that affect the germination, growth, survival, and reproduction of its competing organisms inhabiting the same environment [22]. Allelopathy presents an emerging and effective mechanism for HCB control that has recently attracted attention for its low cost, low toxicity, biodegradability, and environmental friendliness. Quite a few allelochemicals, that is, biochemical compounds produced from the secondary metabolism of macrophytes and microorganisms, have been isolated and identified. A higher efficacy is anticipated if biomanipulation and allelopathy are combined because allelopathic macrophytes can provide shelter for the zooplanktonic grazers. Overall, the greatest advantages of biological methods include high target/host specificity, low energy consumption, and good environmental sustainability.

## 3. Nanotechnology and Nanoparticles

As defined by the U.S. National Nanotechnology Initiative, nanotechnology is the science and engineering of nanoscale matter that demonstrates distinctive phenomena and novel applications (see https://www.nano.gov/about-nanotechnology; accessed on 10 January 2024). Nanoparticles (NPs), defined as particles with at least one dimension in the range of 1 to 100 nm [23], exhibit unusual physical (optical and magnetic), chemical, and biological properties, differing in important ways from those of bulk materials because of their small size approaching the atomic scale and their increased surface area to volume ratio, with a large fraction of the exposed surface lying within a few atomic diameters of its surface [24]. For instance, some NPs are stronger and better at conducting heat or electricity, or they become more chemically reactive, reflect light better, or change color as their size or structure is altered. NP crystals take various shapes, including spherical, rod, oval, needle, triangular, cubic, star, pentagonal, hexagonal, octahedral, flower, branched, platelet, cylinder, and cluster, which enables their application to diverse areas such as device manufacture, electronics, optics, imaging, aerogel, aerospace, automotive, textile, biomedicine, and biofuel cells [25,26,27].

NPs can be classified according to dimension/shape (e.g., 0D quantum dots, 1D nanofiber/wire/tube, 2D nanofilm/layer/disc), phase compositions (single phase solids like crystalline, amorphous particles, and layers, and multi-phase solids like matrix composites, colloids, and ferrofluids), nature (pure metals Au, Ag, and Ni; metallic oxides CuO and TiO_2_; chalcogenides CdS and ZnS; bimetallic or multi-elemental Pt-Ni and CoFe_2_O_4_; and organics such as liposomes and micelles), origin/source (natural NPs originated from storms, dust particles, volcanoes, and microorganisms; anthropogenic nanomaterials such as engineered NPs), and crystallinity (amorphous, crystalline, and polycrystalline) [28,29]. Based on the type of fabrication material used, NPs are categorized into three main types, namely organic NPs, inorganic NPs, and carbon-based NPs. Organic nanoparticles are further classified into polymeric NPs, lipid-based NPs, viral NPs, and protein-based NPs, whereas inorganic NPs consist of metal NPs, silica NPs, magnetic NPs, and quantum dots. Further, carbon-based NPs include carbon nanotubes, graphene, and fullerenes.

According to a market analysis (see https://www.rootsanalysis.com/reports/nanoparticle-formulation-market.html; accessed on 10 January 2024), the global NP formulation market was estimated to be worth $5.1 billion in 2023, with an annual production of 11 million metric tons. NP production presently relies on physical and chemical synthesis methods (e.g., hydrothermal, sonochemical, and laser ablation), which, however, are energy intensive, require high temperatures (60–950 °C) and pressure (~1000 bar), and produce toxic by-products [25,30]. As a more cost- and energy-efficient, safer, greener, and environmentally friendly alternative, a wide variety of organisms (plants, algae, fungi, (cyano)bacteria, and viruses) and cell extracts have been used to synthesize NPs via intracellular and extracellular electron transport systems (e.g., energy-generating reactions in photosynthesis and NADPH-dependent reductase and redox reactions) [25,31,32,33]. Biosynthesis-based green nanotechnology can decrease the consumption of energy and non-renewable raw materials, produce less greenhouse gas emissions and other waste, and reduce potential environmental and human health hazards associated with NP production [34]. Biosynthesis may replace physical and chemical methods if it can overcome such drawbacks as high costs and low scalability.

In the remaining sections of this review, we focus on the emerging approach of nanoparticle application to HCB control, its underlying toxicological mechanisms, existing challenges and limitations, and prospective directions.

## 4. Emerging Application of Nanoparticles to HCB Mitigation

With the rapid advancement of nanotechnology, more varieties of nanomaterials have been fabricated and used in a wider range of products, including common household cosmetics (e.g., nano-TiO_2_- and nano-ZnO-containing sunscreen), appliances, cleaning agents, clothes, tableware, and children’s toys [35]. As one of the emerging applications, four types of NPs (based on chemical composition) have been reportedly tested for treating HCBs: (1) inorganic metal (e.g., Au/Ag/Al/Cu)- or metal oxide (ZnO/CeO_2_/Fe_3_O_4_)-based NPs, (2) carbon-based NPs (graphene, fullerene, single- or multi-walled carbon nanotube (SWCNT or MWCNT), and carbon nanodot (CND)), (3) organic (excluding carbon)-based NPs (dendrimer, cyclodextrin, liposome, and micelle), and (4) composite-based NPs (any combination of types (1) to (3) NPs forming complicated structures like metal–organic framework or MOF) [36,37].

### 4.1. NPs as Algaecide: From Cytotoxicity to HCB Mitigation

The fast-growing and diversified applications of NPs have inevitably led to increased release of manufactured NPs into the environment through discharge of production wastes and disposal of NP-containing products. Consequently, the potential adverse environmental impacts of NPs have raised public and regulatory concerns. Researchers have investigated the biological effects of NPs on cyanobacteria over the past two decades. Numerous studies have demonstrated that NPs may adversely affect gene expression, cellular metabolism, photosynthesis, nitrogen fixation, pigment and protein contents, enzyme activity, and growth rate in exposed cyanobacteria, with the main underlying mechanisms being oxidative stress, mechanical damage, light shielding, and the toxicity of metal ions released from NPs into water bodies [35,37]. The cytotoxic effects of NPs are dependent on their morphology, size, chemical composition, concentration, solubility, and dispersion, as well as cyanobacterial species, cell shape/structure, physiology, and physiochemical characteristics [35]. For instance, Xu et al. [38] found that 72 h exposure to 10 nm TiO_2_ NPs induced more severe cellular damage to a cyanobacterium, *Synechocystis* sp., compared to 50 nm TiO_2_ NPs. The 10 nm TiO_2_ NPs caused significant growth and photosynthesis inhibition in *Synechocystis* sp. cells, as reflected by decreased growth rate (38%), operational PSII (photosystem II) quantum yields (40%), phycocyanin (51%), and allophycocyanin (63%), and increased ROS (reactive oxygen species) content (245%) and SOD (superoxide dismutase) activity (46%) [38]. Transcriptomic analysis of *Synechocystis* sp. Exposure to 10 nm TiO_2_ NPs showed up-regulation of D1 and D2 protein genes (*psbA* and *psbD*), ferredoxin gene (*petF*), and F-type ATPase genes (e.g., *atpB*), and down-regulation of *psbM* and *psb28-2* in PSII [38]. In another study of 96 h exposure to nano-TiO_2_ (10 nm), Cherchi and Gu [39] observed significant inhibition of both growth and nitrogen fixation rates with an EC_50_ of 0.62 mg/L in the exposed cyanobacterium *Anabaena variabilis*. Longer exposure (up to 21 days) resulted in abnormal changes in intracellular C:N, C:P, and N:P stoichiometries [40]. Our own toxicological studies [41] demonstrated that SWCNT and CND both showed adverse effects after 48 h exposure to at least one cyanobacterial strain (e.g., *M. aeruginosa*) and that their toxicity was species- and concentration-dependent. SWCNTs were more toxic than CND, as SWCNTs significantly decreased the turbidity of seven representative cyanobacterial species by 22–95% at a lower concentration [41].

Prior to eliciting any toxic effects, NPs need to interact with and internalize cyanobacterial cells. Cell-NP interactions could reduce light availability, damage cell membrane integrity, and present an obstacle to substance exchanges between the cell and its surrounding environment [42,43,44]. Such interactions may result in the attachment of NPs (e.g., CNTs, SWCNTs, MWCNTs, graphene, graphene oxide, nano-Al_2_O_3_, nano-CuO, nano-TiO_2_, nano-Au, nano-Ag, and carbon/CdSe/CdTe/CdS/ZnS/CuInS_2_ quantum dots) to the surface of the cells to absorb or block part of the light, inducing light-shielding effects that further suppress photosynthesis, growth, and reproduction [37,45,46,47,48]. In the Xu et al. study [38], SEM images indicated that the *Synechocystis* sp. cell surface was heavily entrapped by TiO_2_ NPs, leading to a shrinkage of cell morphology and compelling evidence of cell membrane damage and plasmolysis [42,49].

Consequently, these lines of evidence for cyanotoxicity, along with similar ones for microbial toxicity and phytoplankton toxicity, have led to the beneficial application of NPs to HCB mitigation.

### 4.2. NPs as Photocatalysts for HCB Mitigation

What makes NPs an attractive and promising algicidal agent for HCB mitigation is their unique and extraordinary photocatalytic properties. Photocatalysis (=photon + catalysis) is a type of chemical reaction that involves the absorption of light by one or more reacting species through the addition of substances (catalysts) that participate in the chemical reaction without being consumed. Upon exposure to photons, the electrons in NPs (as a nanophotocatalyst) undergo excitation and transition from the valence band (VB) to the conduction band (CB), thereby creating electron vacancies (holes) within the valence band that can react with other compounds to produce free radical species or reactive oxygen species (ROS) [50]. Such produced intracellular free ROS include superoxide radicals (O_2_^−^), single oxygen (^1^O_2_), hydroxyl radicals (•OH), and hydrogen peroxide (H_2_O_2_) [51]. It has been proven that the production of ROS can damage intracellular lipids, carbohydrates, proteins, DNA, and other biomacromolecules, leading to inflammation, oxidative stress, and oxidation–reduction imbalance in cells [35,47,52]. Oxidative stress refers to a cellular status with a variety of harmful stimuli, including high activity of free radicals and an imbalance of the oxidation system and the antioxidant defense system, which eventually leads to organelle damage and cell death [53]. Furthermore, NP-based photocatalysis provides a pathway for cyanotoxin degradation. As a secondary product of cyanobacteria metabolism, MC-LR (microcystin-leucine arginine) is one of the most harmful cyanotoxins found in water bodies [54]. It has been determined that MC-LR degradation occurs at four sites of the molecular structure (Figure 1): the aromatic ring, the methoxy group, the conjugated double bond of the Adda group, and the cyclic structure of the Mdha amino acid [55]. The conjugated double bond and the methoxy group in Adda and the conjugated system in Mdha are liable to be attacked by hydroxyl radicals (•OH) released by the nanophotocatalyst and converted to a HO-C-OH structure with further oxidation [56]. The formed aromatic ring and double bonds will be further oxidized to another substance or directly mineralized. After the breakdown of the conjugated system in Mdha, the carboxyl group and amino group of the peptides are hydrolyzed. Subsequently, the side chain of the amino acid could be oxidized and mineralized. The Adda chain is deduced to be destroyed and separated from the heptapeptides to produce alkyl derivatives [57,58].

In the field of photocatalytic catalysts for HCB mitigation, nTiO_2_ has been widely investigated because of its chemical stability, low toxicity, low cost, and high photocatalytic activity, while other types of nanophotocatalysts (e.g., ZnO, AgBiO_3_, etc.) have emerged [4]. For example, a nanophotocatalytic TiO_2_ effectively destructed *M. aeruginosa* [59] and broke down intracellular and extracellular microcystins [59,60].

NPs possess great advantages over traditional photocatalytic materials due to the large specific surface areas of NPs. However, researchers have developed various strategies, including doping, compounding, recycling, and replacing, to further enhance the efficacy, sustainability, and environmental friendliness of nanophotocatalysis.

#### 4.2.1. Doping and Compounding

Although nTiO_2_ has a high photocatalytic activity with UV lights (<388 nm) due to its wide band gap (e.g., Eg ≈ 3.2 eV for the anatase TiO_2_ phase), it has a very limited utility in the visible light irradiation in the abundant natural light. The doping strategy addresses this drawback by utilizing such dopants as elements C, N, S, and F to narrow the band gap or form an intraband gap of TiO_2_ NPs and decrease the required activation energy [61]. For example, N-doping can narrow the band gap through substitution lattice sites by mixing N2p with O2p states in the valence band [62]. In S-doping, the overlap of S3p and O2p states facilitates the visible light catalytic activity of S-doped TiO_2_. For low concentrations of C-doping, C atoms prefer to be interstitial and substitutional to Ti atoms under oxygen-rich conditions or substitutional to O under anoxic conditions [63,64].

Doping enables the precise design of photocatalytic NPs tailored to specific bandgaps, energy levels, and surface activity to improve the efficacy and capability of producing ROS. For example, doping non-metal elements N-F, S, and C decreased the band gap (~3.2 eV) of the anatase TiO_2_ phase to 2.9, 2.8, and 2.7 eV, respectively, while the photocatalytic properties of the doped nTiO_2_ were significantly improved under visible light [4,65]. Wang et al. [66] doped N and P into the nTiO_2_ crystal by controlling the calcination temperature, and the doped nTiO_2_ exhibited improved performance of 81.5% mortality for *M. aeruginosa* following a 6 h visible light irradiation. Doped nTiO_2_ demonstrated higher photodegradation rates of MC-LR owing to: (a) high adsorption rates of doped-TiO_2_, leading to a higher photocatalytic potential; (b) electrons promoted from the VB to the CB, leading to the formation of energized holes on the surface of the TiO_2_; and (c) electron–hole recombination obstructed by doped elements, leading to enhanced efficacy of photocatalytic degradation [4]. Therefore, element doping could be attributed to a shift of the energy band gap to the visible range (<3.2 eV), activating visible light photocatalysis during MC-LR degradation [67,68].

An alternative strategy to overcome the low activity of nTiO_2_ or other metal-based NPs under visible light is to compound them with such NPs as graphitic carbon nitride g-C_3_N_4_ and terephthalic acid-functionalized g-C_3_N_4_ (TACN) that can be excited under visible light. For instance, heterojunction composites TiO_2_/g-C_3_N_4_ [69,70], Bi-TiO_2_/g-C_3_N_4_ [71], Ag_2_MoO_4_/TACN [72], WO_3_/g-C_3_N_4_ [73], SnO_2_/g-C_3_N_4_ [74], and CdS/g-C_3_N_4_ [75] displayed enhanced photocatalytic performance under natural sunlight.

Recently, metal–organic frameworks (MOFs) have attracted significant attention because they are highly versatile NPs composed of inorganic nodes interconnected by organic linkers that form a porous crystalline structure. MOFs possess high photon capture efficiency, a large specific surface area, and adjustable porosity, making them highly effective photocatalytic NPs. As summarized by Song et al. [50], the following two fabricated MOFs were used to treat *M. aeruginosa* at doses as low as 6 mg/L with >90% removal rates in as short as 4 h: Ag/AgCl@ZIF-8 [76] and g-C_3_N_4_/Cu-MOF [77].

#### 4.2.2. Recycling and Replacing

The recycling strategy refers to reactivating photocatalytic activity and reusing the NPs, whereas the replacing strategy means replacing metals with organic non-metallic elements in the NPs. These two strategies address the sustainability and environmental friendliness issue, that is, the avoidance of secondary contamination by metal NPs if released into the environment. It was reported that metal-containing NPs (e.g., nAg, nZnO, nPbS, and nCu_2_O) would gradually dissolve and release metal ions through various physiochemical and biological processes in the environment, and the dissolved metal ions may induce toxic effects on non-target phytoplankton and zooplankton [78,79,80,81]. In addition, the NP uptake by cyanobacteria and other low-trophic species may be accumulated in higher trophic organisms through the food chain. For instance, it was reported that more than 70% of nAg was accumulated in *Daphnia magna* through the ingestion of algae [82].

To implement the replacement strategy, metal-free semiconductor nanophotocatalysts such as g-C_3_N_4_ have been developed. They have the advantages of low cost, high stability, and non-toxicity, and they could improve charge carrier separation efficiency [83,84]. Furthermore, g-C_3_N_4_ exhibits a high photocatalytic response under visible light due to its low band gap (2.7 eV) [74,75]. However, g-C_3_N_4_ has a high electron–hole pair recombination rate, which may impair its photocatalytic performance [73,85]. Such modified g-C_3_N_4_ as TACN can enable photoexcited e^−^ aggregation on O-containing chains and h^+^ aggregation on N-containing chains, which jointly slow down the recombination of e^−^ and h^+^ and ultimately improve the photocatalytic performance [69].

Abiding by the recycling strategy, several researchers turned powder-formed NPs into conveniently recoverable structures. For instance, Kennedy et al. [60] immobilized photocatalytic nTiO_2_ in thermoplastic polymer composites that can be 3D-printed as customizable, high-surface-area deployable, retrievable, and reusable geometric lattices that degraded 50% of MC-LR in 3 h. Fan et al. [86] prepared and loaded a composite heterojunction nanophotocatalyst Ag_2_MoO_4_/TACN onto loofah (a layered, multi-mesh, and porous sponge) using an oscillating impregnation method to form a floating photocatalyst that achieved 100% chlorophyll removal within 4 h of visible light irradiation. A simple facile sol–gel technique was employed to develop honeycomb-like hetero-structures of floating nanophotocatalysts, that is, F-Ce co-doped TiO_2_ distributed on an expanded perlite (EP) surface [87] or coating g-C_3_N_4_ with Bi-doped TiO_2_ on Al_2_O_3_-modified EP [71], both of which showed enhanced photocatalytic inactivation of *M. aeruginosa* under visible light. Qi et al. [88] synthesized a recyclable magnetic Zn-doped Fe_3_O_4_ visible-light catalyst whose efficiency only slightly decreased after three regeneration cycles. Fan et al. [72] fabricated a ternary nanocomposite ZnFe_2_O_4_/Ag_3_PO_4_/g-C_3_N_4_ by taking advantage of the strong paramagnetism of ZnFe_2_O_4_ and its narrow bandgap, low toxicity, and high photocatalytic stability. The magnetic property facilitates in situ separation, recovery, and reuse of nanophotocatalysts.

### 4.3. NPs as Flocculant/Coagulant for Cyanobacteria Removal

As a chemical method for HCB mitigation, traditional metals (e.g., Al, Fe, and Ti) and modified clay-based coagulants can aggregate and sediment cyanobacterial cells through electrostatic adsorption without damage to cell membrane integrity and the consequent release of cyanotoxins [89]. However, the performance is limited by the negative charge on the surface of cyanobacterial cells. In contrast, nanocationic coagulants offer superior flocculation performance owing to their high positive charge density, large specific surface area, and enhanced adsorption and bridging effects [50]. For example, a magnetic composite flocculant Fe_3_O_4_/CPAM (cationic polyacrylamide) achieved a 97% reduction of chlorophyll a in algae-laden raw water at an extremely low dosage of 1.2 mg/L within 9 min [90]. The magnetic property of this nanoflocculant also facilitates recovery and recyclability. In another study, a novel strategy combining palladium clusters (Pd_n_) with g-C_3_N_4_ nanocomposite showed a highly efficient 95% removal of *M. aeruginosa* cells with an initial density of 5.6 × 10^6^ cells/mL within 10 min in the dark through coagulation and breakage [91]. Li et al. [92] demonstrated that the water-stable Cr(III)-based MOFs, structured as NH_2_-MIL-101, could be used for the removal of *M. aeruginosa* by coagulation/flocculation with >95% efficiency within 1.5 h at a 30 mg/L dosage or 3 h at a 20 mg/L dosage over a wide range of pH and cell densities.

Such natural coagulants as chitosan, a non-toxic and biodegradable organic polymer derived by the simple alkaline deacetylation of chitin, also garnered increasing attention due to their biodegradability, environmental friendliness, cost effectiveness, and high efficiency [50]. A major source of chitin is the leftover shells of abundant crustacean seafood, such as shrimp, prawns, crabs, and lobsters. For example, Chen et al. [93] synthesized an amphoteric chitosan-based flocculant CPCTS-g-P (CTA-DMDAAC) by UV-initiated graft copolymerization using carboxylated chitosan (CPCTS), 3-chloro-2-chloropropyltrimethylammonium chloride (CTA), and dimethyldiallylammonium chloride (DMDAAC) as the cationic co-monomers. Results from flocculation experiments showed a 98.8% removal rate of *M. aeruginosa* measured as chlorophyll a content in 20 min at a low dosage of 4 mg/L [93].

The nanocoagulants not only flocculate cyanobacterial cells but also the dissolved or bound extracellular organic matter (EOM) that contains humic acid-like substances, tryptophan-like proteins, and metabolites like MC-LR [90,94]. For instance, the Fe_3_O_4_/CPAM nanocomposite flocculant removed 87% EOM by binding with the functional groups in tryptophan-like proteins, such as amino, carboxyl, and hydroxyl groups [90], whereas the efficiency of co-graft tannin (TA)-based flocculants, TA-g-P(AM-DMDAAC), in the removal of *M. aeruginosa* cells, EOM, and MC-LR increased with the increase in the charge density and molecular weight of the flocculants [94].

To overcome the drawbacks of settling flocculants (i.e., reliance on complex and expensive equipment for filtration and aeration devices required for flotation [95]), various novel flotation technologies have been developed to enable one-step removal of bloom-forming cyanobacteria. For example, Lin et al. [96] utilized a novel flocculant (self-branched chitosan) integrated with flotation function (induced by CaO_2_@PEG) to develop a CP-SBC (CaO_2_@PEG-loaded water-soluble self-branched chitosan) system that removed multiple algae species from water in one step without additional instrumentation.

### 4.4. NPs as an Adsorbent for Nutrient and Cyanotoxin Removal

Removal of nutrients, especially phosphorus, from bloom water by adsorption is a commonly employed and indirect approach to controlling HCBs. Unlike traditional adsorbents, NPs are characterized by their large specific surface areas and surface energy, abundant pores, layered structures, and the distribution and variety of elements in the structures, resulting in their exceptionally strong adsorption capacity [50]. For example, although bulk lanthanum (La)-based adsorbents (e.g., La_2_O_3_ and La-modified bentonite) possess > 10 times greater affinity and capacity of phosphorus (P) binding and are more effective in P precipitation over a wider pH range (4.5 to 8.5) than other metal-based (e.g., Al, Fe, Ti-based) adsorbents [97], nanosized La-based adsorbents have displayed even higher efficiency in P removal [4]. Based on the properties and types of carriers, the nanosized La-based adsorbents can be classified into five categories (% of total): La-clay minerals (48.2%), La-organics (20.6%), La-metallic compounds (10.6%), La-silica (2.3%), and La-other substances (18.3%) [4]. The thermodynamic kinetics of the P adsorption process on these nanomaterials fits a pseudo-second-order model, with underlying mechanisms governed by ion exchange, surface co-precipitation, electrostatic attraction, and Lewis acid–base reactions [4,98]. The P adsorption process is regulated by such factors as the pH, P/La ratio, coexisting anions, temperature, and sorption conditions [4]. Zhang et al. [98] synthesized a La(OH)_3_ nanoadsorbent with an ultrahigh P adsorption efficiency, and Chen et al. [99] loaded La(OH)_3_ into magnetic mesoporous silica nanospheres that showed a superior pH (4 to 11) stability, highly efficient P removal properties, and ease in NP separation and recovery.

Recently, nanozero-valent iron (nZVI) has attracted widespread attention as an excellent nanoadsorbent owing to its low cost, non-toxicity, and high specific surface area [50]. The primary mechanism involved in P removal by nZVI is chemisorption, in addition to the possibility of physical–electrostatic deposition of P-species onto the surface of NZVI within the liquid–solid controlling step: liquid film and liquid–solid diffusion [100]. For example, Zhou et al. [101] prepared a nZVI-loaded sugarcane bagasse (nZVI/SCB) composite with excellent stability (stored stably for 450 days) and high P sorption capacity (205.2 mg/g at a dose of 1600 mg/L) by liquid phase reduction. Shanableh et al. [102] made a chitosan-coated nZVI with a higher adsorption capacity (437 mg/g) at a lower dose of 300 mg/L.

In addition to nutrient removal, NPs have also been used to remove cyanotoxins. Take MC-LR, one of the most common and most toxic variants of isolated or described microcystins (MCs), as an example. The MC-LR molecule is 2.94 nm in length [103] with two substitutions of leucine (L) and arginine (R) at positions 2 and 4 (see Figure 1). MC-LR adsorption to NPs can fit Langmuir equations with pore sizes between 2 and 50 nm for high-capacity nanoadsorbents [4]. The multiple carbonyl and carboxyl groups in MC-LR also have a strong affinity toward the metal atoms in MOFs or other metal-based NPs [104]. In adsorption tests, the N-doped carbon xerogel (N-CX) was efficient for MC-LR adsorption, with an adsorption capacity of 1916 μg/g, which was higher than that of commercial activated carbon (1034 μg/g) and graphene oxide (1700 μg/g) [57]. N-CX was recyclable after desorption treatment by washing with NaOH solution, with no loss of adsorption capacity within five cycles [57].

## 5. Challenges, Limitations, and Prospectives

The unique physical and chemical properties of NPs have enabled their versatile applications, including HCB mitigation. The past decade, especially the last 5 years, has witnessed an explosion in R&D activities on innovative NP-based technologies for HCB management. As described above, both manufactured and modified natural NPs can act as algaecide (cyanocide), photocatalyst, coagulant/flocculant, or adsorbent to directly remove or deactivate cyanobacterial cells and cyanotoxins or indirectly reduce/recover nutrients (P) from eutrophic bloom water (Figure 2). Compared with conventional chemical and physical methods, NP-based approaches offer great benefits in terms of environmental friendliness, effectiveness, and sustainability [4,50]. Many of the recently developed NP-based technologies hold great promise for field deployment to remediate HCB in real-world environments.

The main challenges we face include high costs associated with NP synthesis or composite fabrication, scalability to field-level operation, and environmental impact assessment for future approval by government regulatory agencies. So far, all reported studies are limited to laboratory and micro-/mesocosm experiments under controlled environmental conditions. The used NPs were synthesized or fabricated by in-house researchers. To reduce costs, the nanomaterials need to be commercialized and mass produced through mechanized manufacturing processes.

As for technological scalability, further research is warranted to evaluate if a specific nanotechnology is influenced by abiotic and biotic factors under field conditions. Through such studies, we should be able to identify top-ranked technologies that can maintain consistently superior performance in real-world environments.

To meet regulatory requirements and ensure environmental safety and sustainability, we need to assess the ecological risks of nanotechnologies when applied to mitigation of HCB-infested aquatic ecosystems, especially toxicological effects on those non-target species cohabiting in the same environment (e.g., open lakes, seas, or other water bodies) as the bloom-forming cyanobacteria [105]. In addition, any environmental residues of released NPs, biomagnification potential, and long-term impacts on environmental quality and human health need to be further investigated.

NP-based HCB mitigation technologies are not free of limitations. As a chemical or physical treatment approach, all nanotechnologies have low target species specificity. It is widely recognized that no technology can solve all HCB-associated problems (cyanobacterial biomass, cyanotoxins, eutrophication, and biodiversity). An ideal and comprehensive solution to all these problems would require the use of different NP-based technologies and possibly other biological, chemical, and physical methods, as summarized in the earlier part of this review.

Further efforts may focus on evaluating and scaling up promising NP-based mitigation technologies (e.g., those based on photocatalytic and flocculant/coagulant NPs) from bench- or laboratory-scale to real-world operations through developing more efficient, recoverable, reusable, and deployable NP-based lattices or materials that are adaptable to bloom events in different water bodies of different sizes, such as reservoirs, lakes, rivers, and marine environments.

## Figures and Tables

**Figure 1 toxins-16-00041-f001:**
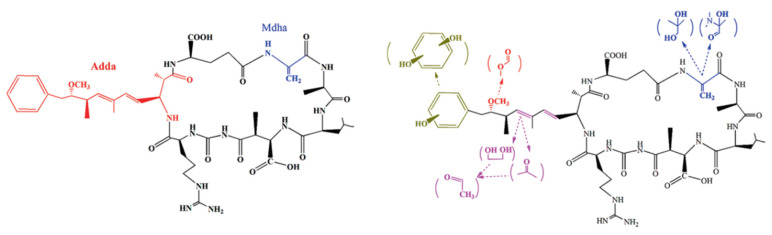
Microcystin (MC)-LR molecular structure (**left**) and proposed photocatalysis degradation pathway (**right**). Adapted from Cheng et al. [4] with permission from Springer Nature.

**Figure 2 toxins-16-00041-f002:**
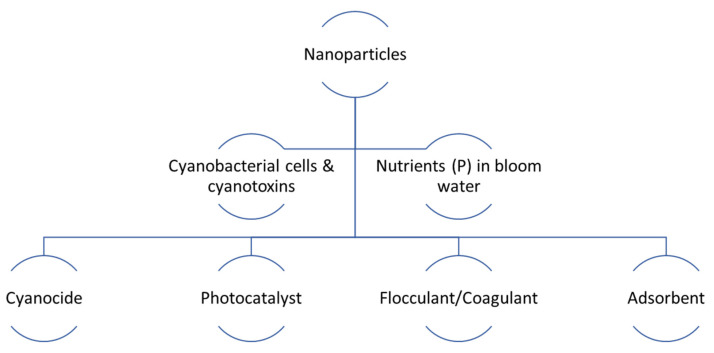
Emerging applications of nanoparticles to harmful cyanobacterial bloom management.

## Data Availability

No new data were created or analyzed in this study. Data sharing is not applicable to this article.

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
