# Peer review of "Nanoparticles for Mitigation of Harmful Cyanobacterial Blooms"

_toxins, 2024, doi:10.3390/toxins16010041_

Round 1
Reviewer 1 Report
Comments and Suggestions for Authors
The present review deals with possible techniques to use to reduce the impact of algal blooms. In previous reviews, which the authors also cite, the various methods studied are discussed, but this manuscript focuses on the possible use of nanoparticles to achieve the same purpose. The manuscript is well written, there are only some small errors and grammatical inaccuracies. However, I ask myself 2 questions that I would like the authors to answer and if possible, report their opinion in the manuscript:
1. Nanoparticles are considered toxic for many organisms, if we consider treating blooms with nanoparticles in the open sea, couldn't there be negative effects for planktonic and non-planktonic species (invertebrates and vertebrates) that share the same environment? The problem has been mentioned by the authors, but their opinion on the matter is not clear.
2. The experiments reported are all on a small scale or in the laboratory. Is it possible to apply the various methods in situations of algal blooms in large water basins (lake or sea)?
Comments (minor revision)
It would be better to specify in the first sentence (paragraph 2.1, line 83) the purpose for which physical and chemical methods were developed (to mitigate HCB effects).
Little information about allelopathy is here reported. How does it work?
Line 168: target/host
Author Response
Reviewer #1
Comments and Suggestions for Authors
The present review deals with possible techniques to use to reduce the impact of algal blooms. In previous reviews, which the authors also cite, the various methods studied are discussed, but this manuscript focuses on the possible use of nanoparticles to achieve the same purpose. The manuscript is well written, there are only some small errors and grammatical inaccuracies. However, I ask myself 2 questions that I would like the authors to answer and if possible, report their opinion in the manuscript:
- Nanoparticles are considered toxic for many organisms, if we consider treating blooms with nanoparticles in the open sea, couldn't there be negative effects for planktonic and non-planktonic species (invertebrates and vertebrates) that share the same environment? The problem has been mentioned by the authors, but their opinion on the matter is not clear.
Response: Toxicological effects of NPs were one of the multiple mechanisms explored for beneficial application of NPs to HCB mitigation. We stated in the last section that toxicity or adverse side effects of NPs on non-target organisms cohabiting the same environment (like open sea, lakes or other water bodies) as bloom-forming cyanobacteria need to be investigated and assessed to ensure environmental safety and sustainability of any NP-based technology. In addition, we pointed out that other ecological impacts should also be evaluated, including environmental NP residues, biomagnification potential and long-term environmental quality and human health. We have revised this paragraph (original lines 531-535) to reinforce our opinion on this matter. See below for the revised paragraph:
“To meet regulatory requirements and ensure environmental safety and sustainability, we need to assess the ecological risks of nanotechnologies when applied to mitigation of HCBs-infested aquatic ecosystems, especially toxicological effects on those non-target species cohabiting the same environment (e.g., open lakes, seas, or other water bodies) as the bloom-forming cyanobacteria. In addition, any environmental residues of released NPs, biomagnification potential, and long-term impacts on environmental quality and human health need to be further investigated.”
- The experiments reported are all on a small scale or in the laboratory. Is it possible to apply the various methods in situations of algal blooms in large water basins (lake or sea)?
Response: Application of NPs to HCBs (or HABs) mitigation has been an active R&D field only for about a decade or so. As pointed out in the very last paragraph, further efforts are required to scale up and evaluate promising technologies (e.g., those NPs acting as photocatalyst, flocculant/coagulant or adsorbent), which do not involve the release of NPs to the environment through recovering, reusing and regenerating NPs. So, our view on this issue is very clear – yes, it is more than possible to apply the NP-based photocatalyst, flocculant/coagulant or adsorbent to treat in situ the HCBs in large-scale water bodies like lakes and coastal seas. See below for the slightly revised last paragraph with
“Further efforts may focus on evaluating and scaling up promising NP-based mitigation technologies (e.g., those based on photocatalytic and flocculant/coagulant NPs) from bench- or laboratory-scale to real-world operations through developing more efficient, recoverable, reusable, and deployable NP-based lattices or materials that are adaptable to bloom events in different water bodies of different sizes such as reservoirs, lakes, rivers, and marine environments.”
Comments (minor revision)
It would be better to specify in the first sentence (paragraph 2.1, line 83) the purpose for which physical and chemical methods were developed (to mitigate HCB effects).
Response: Added “to mitigate HCB effects” to paragraph 2.1 (lines 78-79 in the revised version) as suggested.
Little information about allelopathy is here reported. How does it work?
Response: Added one sentence to explain how allelopathy works. See lines 160-162 for “Allelopathy is a biological phenomenon by which an organism (e.g., plants, bacteria, coral, and fungi) produces allelopathic biochemicals that affect the germination, growth, survival, and reproduction of its competing organisms inhabiting the same environment (Yunes 2019)”.
Line 168: target/host
Response: Changed “targe/host” to “target/host” as suggested.
Reviewer 2 Report
Comments and Suggestions for Authors
This manuscript is gives a short review regarding te use of nanoparticles to mitigate the harmful cyanobacterial blooms which are increasing world wide. The review is very accurate on the subject and it is well write.
In the text, the references should be cited as their reference numbers not by the authors names. See other the suggestions made to improve the manuscript.

The English Language is very good.
Author Response
Reviewer #2
Comments and Suggestions for Authors
This manuscript gives a short review regarding the use of nanoparticles to mitigate the harmful cyanobacterial blooms which are increasing worldwide. The review is very accurate on the subject and it is well write.
In the text, the references should be cited as their reference numbers not by the authors names.
Response: We have reorganized the references and numbered them in the order of appearance in the text (including citations in tables and legends).
Other suggestions in peer-review-33991782.v1.pdf made by the reviewer:
- Comment: HCB is not a red tide (original lines 57-59)
Response: We agree with the reviewer and replaced “For instance, an HCB (or red tide) event wiped out 90% of the entire stock of Hong Kong’s aquaculture in 1998, resulting in an estimated economic loss of $40 million (Ho 2022)” with the following new sentence: “In addition, a 10% increase in HCB frequency reduced the average home values by 3.3~4.3% for properties near > 2000 US inland lakes during the years 2008–2011 [14].”
- Typos of HCH at three locations (original lines 146, 149 and 162)
Response: Replaced all three HCH with HCB.
- Suggested change: cellar shape/structure (original line 245)
Response: This is a misspelling. We have corrected “cellar” to “cell”.
- Suggested change: For instance, Xu et al. (2021) investigated toxic effects of two sizes of TiO2 NPs (50 and 10 nm) on a cyanobacterium, Synechocystis sp. and found that 72-hr exposure to 10-nm TiO2 NPs induced more severe cell damages compared to 50-nm TiO2 NPs on Synechocystis sp. (original lines 246-249).
Response: Accepted the reviewer’s suggestion and changed the sentence to “For instance, Xu et al. (2021) found that 72-hr exposure to 10-nm TiO2 NPs induced more severe cellular damages to a cyanobacterium, Synechocystis sp., compared to 50-nm TiO2 NPs.”
- Comment: excited electron? Comment: hydrogen ion? (original line 393)
Response: Yes, the reviewer was right about both e- and h+.
- Suggested change: without damage to cell membrane integrity (original line 417)
Response: We did not follow the reviewer’s suggestion because we think “damage to” is an appropriate expression.
- Suggested change: aglaecide (cyanocide) (original line 511)
Response: There is a typo with “aglaecide” which should have been “algaecide”. We did not follow the reviewer’s suggestion to remove “algaecide” because it is a more commonly accepted term than the newly coined “cyanocide” and it has a broader scope covering both cyanobacteria and non-cyanobacterial planktonic algae. We think it is more appropriate to use algaecide along with a bracketed phrase of cyanocide to specify the scope of coverage.
Comments on the Quality of English Language
The English Language is very good.
Response: We very much appreciate the complimentary comment.